# The Interaction of Seasons and Biogeochemical Properties of Water Regulate the Air–Water CO_2_ Exchanges in Two Major Tropical Estuaries, Bay of Bengal (India)

**DOI:** 10.3390/life12101536

**Published:** 2022-10-02

**Authors:** Suchismita Pattanaik, Pradipta Kumar Mohapatra, Debasish Mohapatra, Sanhita Swain, Chitta Ranjan Panda, Pradeep Kumar Dash

**Affiliations:** 1Council of Scientific and Industrial Research (CSIR)—Institute of Minerals and Materials Technology, Bhubaneswar 751013, India; 2Department of Botany, Ravenshaw University, Cuttack 753003, India; 3Indian Council of Agricultural Research (ICAR)—National Rice Research Institute, Cuttack 753006, India

**Keywords:** estuary, air–water CO_2_ flux, physicochemical parameters, biological parameters, chlorophyll fluorescence, Bay of Bengal

## Abstract

The exchange of CO_2_ between the air–water interfaces of estuaries is crucial from the perspective of the global carbon cycle and climate change feedback. In this regard, we evaluated the air–water CO_2_ exchanges in two major estuaries—the Mahanadi estuary (ME) and the Dhamra estuary (DE) in the northern part of the Bay of Bengal, India. Biogeochemical properties of these estuarine waters were quantified in three distinct seasons, namely, pre-monsoon (March to May), monsoon (June to October), and post-monsoon (November to February). The significant properties of water, such as the water temperature, pH, salinity, nutrients, dissolved oxygen, chlorophyll-a (chl *a*), and photosynthetic pigment fluorescence of phytoplankton, were estimated and correlated with CO_2_ fluxes. We found that the ME acted as a source of CO_2_ fluxes in the monsoon and post-monsoon, while DE acted as a sink during the monsoon. The stepwise regression model showed that the fluxes were primarily driven by water temperature, pH, and salinity, and they correlated well with the phytoplankton characteristics. The chl *a* content, fluorescence yield, and phycobilisomes-to-photosystem II fluorescence ratios were major drivers of the fluxes. Therefore, for predicting air–water CO_2_ exchanges precisely in a large area over a seasonal and annual scale in the estuaries of the Bay of Bengal, India, critical key parameters such as water temperature, pH, salinity, chl *a*, and fluorescence yield of phytoplankton should be taken into consideration. However, the responses of phytoplankton, both in terms of production and CO_2_ capture, are critical research areas for a better understanding of air–water CO_2_ exchanges in coastal ecology under climate change scenarios.

## 1. Introduction

A significant component of the global carbon cycle is the supersaturation of carbon dioxide (CO_2_) in estuaries and the resulting efflux to the atmosphere. In biogeochemically active zones such as estuaries, dissolved and particulate materials undergo rapid changes. Estuaries are primarily net heterotrophic, contributing significantly to CO_2_ emissions [1,2,3]. On the other hand, some estuaries occasionally exhibit net autotrophy and act as CO_2_ sinks [4,5]. The physical forcing, the balance between carbon fixation and respiration, and the lateral inputs of dissolved inorganic carbon (DIC), together with heterotrophy and autotrophy, are the major factors that influence the direction and magnitude of estuarine air–water CO_2_ exchanges [6,7]. The generation of CO_2_ in the estuarine water column and the pattern of several biogeochemical processes depend on several variables related to each estuary’s geophysical characteristics [8]. The geomorphology, structure, and function of estuaries worldwide have seen a significant transformation over the past century as a result of anthropogenic interventions as well as climate change.

Globally, estuaries account for 0.27 ± 0.23 Pg C yr^−^^1^, approximately 17% and 37% of the total CO_2_ uptake by the open ocean and continental shelves, respectively [9]. However, a significant uncertainty exists in flux estimation worldwide due to the exclusion of several small-scale estuaries that have been scarcely studied. Therefore, all small and big estuaries should be considered and studied on a spatial and temporal scale to provide a precise flux assessment and develop our understanding of the estuarine CO_2_ budget. However, determining the spatial and temporal fluctuations of surface pCO_2_ of estuarine water and CO_2_ efflux at the air–water interfaces is difficult due to the rapidly changing nature of coastal environments [10]. Many biological, physical, and physicochemical factors drive the CO_2_ fluxes in a particular estuary. Therefore, studying the related physicochemical and biological factors to describe the CO_2_ fluxes is crucial. The biogeochemical properties such as water temperature, pH, salinity, nutrients, and dissolved oxygen, etc., play a key role in the air–water CO_2_ exchanges in the estuary systems of the world [11,12]. To draw firm conclusions on the role that estuaries play in the global carbon budget, a database of small and big estuaries across the globe is necessary.

The Mahanadi estuary (ME) and the Dhamra estuary (DE) are two of the most prominent estuaries on the east coast of India. It is reported that physiochemical processes play a vital role in carbon dynamics in the Mahanadi estuary (ME) and the Dhamra estuary (DE), which regulates CO_2_ fluxes [13,14]. However, the impact of biological variables and photosynthetic efficiency on CO_2_ dynamics, and the inter-relationship between seasonality and biogeochemical property in the ecology, need to be studied. Biological variables could affect CO_2_ exchanges significantly in estuaries [15,16]. Among the biological properties, the fluorescence parameters such as fluorescence minimum (F_0_), fluorescence maximum (F_M_), variable fluorescence, and photosynthetic yield, in particular, have been found to drive the CO_2_ fixation rates in the different coastal and estuarine systems [17,18,19]. Therefore, the specific objective of this study is to assess the impact of seasons and the significant biogeochemical properties of water on air–water CO_2_ exchanges, and the relative importance of the key properties on CO_2_ exchanges in the ME and DE of the Bay of Bengal.

## 2. Materials and Methods

### 2.1. Study Area

The research covers two estuaries on the east coast of India: (1) the ME and (2) the DE, the confluence of rivers Brahmani and Baitarani. The ME and DE are two distinctive ecosystems because they are subjected to extreme environmental conditions and, therefore, are very susceptible to climate change and other anthropogenic interferences. The Mahanadi River is the third largest river in India and the largest river in Odisha, with an annual discharge of 66,640 mm^3^. The basin is marked by a tropical climate, with a mean annual rainfall of 1572 mm, of which 90% occurs during the southwest monsoon. The river starts from the Baster hills of Madhya Pradesh, flowing over different regions of the Eastern Ghats and adjacent areas, and merges into the Bay of Bengal. The main branches of Mahanadi join the Bay of Bengal at Paradip (Mahanadi estuary) and Nuagarh (Devi estuary). ME is a microtidal estuary located between 19°40′–20°35′ N and 85°40′–86°45′ E that receives effluents from fertilizer plants, various other industries, agricultural lands, and domestic sectors from different cities along its bank [20]. The source of pollution also includes the port activity, wastes from the Paradeep port city, and the effluents from Paradeep Phosphates Ltd. The river is a major water supply source for irrigating 13,590 km^2^ of agricultural land in the basin [21]. Out of many ecological features, the most important is the mangrove patch that extends from the southeastern part of Mahanadi river to the estuary point of Hansua (a tributary of Brahmani) in the north and from the northeastern end of Mahanadi river up to Jamboo river in the east.

The tropical DE (20°35′ to 20°50′ N and 86°46′ to 87°05′ E) receives an annual rainfall of about 1670 mm, mainly during August and September (monsoon season). The DE receives water from two major rivers, viz., Brahmani and Baitarani. The pollutant source of DE is primarily contributed by coal mining activities in the Talcher area, chromite in Sukinda, and iron/manganese in Keonjhar district, along with contributions from industrial clusters of Rourkela, Dhenkanal, and Jajpur districts of Odisha, India. During the rainy season, DE receives toxic pollutants, mostly heavy metals, from mining areas and mineral-rich catchment areas. Current activities of Dhamra port, such as dredging, also impact the estuary’s water and sediment quality.

This selected study area has great significance due to the presence of India’s second largest mangrove area of Bhitarkanika wildlife sanctuary, Gahirmatha marine sanctuary (nesting ground of olive ridley turtle), and Dhamra Port [22,23,24]. Considering its social and ecological importance, the Bhitarkanika area has been identified as a Ramsar site [25]. The estuary area is characterized by the presence of small scattered rivulets, creeks, and channels continuously influenced by tides. Both estuaries are regularly affected by tropical cyclones in the Bay of Bengal.

Nine fixed stations from the riverside (Stations 1–3 as upper reach, 4–6 as middle reach, and 7–9 as lower reach) towards the estuary point, maintaining a distance of 1 km between two stations, were selected as sampling points for the collection of water samples, which have been mentioned as estuary stations in both the Mahanadi and Dhamra mouths (Figure 1).

### 2.2. Sampling Strategy

Surface water samples (100 cm from the surface) were collected in duplicates from each station of the ME and DE with the help of a Niskin sampler covering three seasons, namely, pre-monsoon (March to May), monsoon (June to October), and post-monsoon (November to February). For water sampling, pre-cleaned glass bottles were used [for dissolved oxygen (DO)] and polythene bottles (for nutrients, chl *a*, and pigment fluorescence). Sampled water was preserved and analyzed following standard procedures [26].

### 2.3. Biogeochemical Properties

#### 2.3.1. Physicochemical Properties

Water temperature (WT), pH, and salinity were measured immediately after collecting the sample on board by a multi kit (WTW Multi 340 i Set, Germany) WTW Tetracon 325 probe fitted for the analysis of salinity and WT, and WTW Sentix 41-3 probe fitted for pH. The glass electrode for pH measurements was calibrated before sampling in the NBS scale with technical buffers of pH 4.01 (Model TEP Trace at a controlled temperature of 25 °C). Dissolved oxygen content was measured by Winkler’s titrimetric method (accuracy of ±0.1%). Dissolved inorganic nutrients such as nitrate (accuracy of ±2.9%), nitrite, ammonia, and phosphate (accuracy of ±2.5%) were determined by spectrophotometric methods using a bio UV-visible spectrophotometer (Varian 50) [27]. Total alkalinity (TA) was measured potentiometrically using Metrohm automatic titrator inbuilt with Tiamo software (Metrohm 905 Titrando).

#### 2.3.2. Biological Properties

Chl *a* in water samples was estimated by filtering 1000 mL of a water sample using Whatman GF/F (47 mm diameter) under vacuum pressure [28]. The filter was mixed with acetone (10 mL; Merck) and incubated for 24 h at −4 °C; after centrifuging for 15 min at 5000 rpm, the absorbance of the supernatant was measured by spectrophotometer (Varian 50 bio UV-visible). The OJIP fluorescence transient was measured after concentrating the samples and providing a saturating pulse of red light (3000 µE/m^2^s) by a Handy photosynthetic efficiency analyzer (Handy PEA, Hansatech Instruments, UK). For this, 10 mL of water sample was centrifuged, and the pellet with 2 mL of supernatant was put in a 2 mL glass vial with an aluminum screw cap that was kept in the dark at room temperature for 15 min for complete relaxation of PS II reaction centers. Sample vials were put into the liquid attachment of Handy PEA. Ground level fluorescence minimum (F_0_), maximum fluorescence (F_M_), and the variable fluorescence (F_V_ = F_M_ − F_0_) were recorded after applying a saturating light pulse (3000 µE/m^2^s, duration 1 s) [29]. The photosynthetic pigment fluorescence was also measured spectrofluorimetrically using a Shimadzu spectrofluorimeter (Model RF-5301PC, Shimadzu, Japan). Ten mL of the sample was centrifuged, and the pellet with 2 mL of homogenized supernatant was poured into a quartz cuvette (3 mL). The latter was placed in the cuvette holder of the equipment, excited with a 440 nm monochromatic light beam, and the fluorescence emission spectrum was measured from 600 nm to 750 nm. The peak fluorescence at 685 nm was recorded as an emission from PS II. Similarly, the sample was excited with a 570 nm monochromatic beam, and the emission spectrum was measured from 600 to 750 nm. The peak fluorescence at 655 nm was recorded as an emission from phycobilisome (PBS). In all cases, the excitation and emission bandpasses were 10 nm each to optimize the fluorescence signal.

#### 2.3.3. Air–Water CO_2_ Flux

The data of observed salinity, temperature, pH, nutrients (phosphate and silicate), and TA were used to calculate the CO_2_ fugacity of water [*f*CO_2_ (water)] and dissolved inorganic carbon (DIC) using CO2SYS.EXE software [30]. The dissociation constants K1 and K2 were used according to Peng et al. (1987) [31]. The concentration of CO_2_ in the immediate surrounding atmosphere, 10 m above the water surface, was measured by a non-dispersive infrared gas analyzer (Li-840A CO_2_/H_2_O gas analyzer, Li-COR Inc., Lincoln, NE, USA). Both wind speed and ambient CO_2_ concentration were measured above 10 m of the water surface by mounting a portable air pump (for CO_2_ concentration) and an anemometer (for wind speed) with the help of a 7 m-long pole. The measured CO_2_ in ppm was converted to pCO_2_ (air) using the equation of state [32].

Air–water CO_2_ flux densities (µmol/m^2^.h), or FCO_2,_ were estimated by the following equation:
(1)$$FCO_{2}=k\cdot \beta \cdot \Delta fCO_{2}$$
where ‘*k*’ is the gas transfer velocity (cm h^−1^), ‘*β*’ is the Ostwald dilution coefficient (mol m^−3^ atm), and ∆*fCO*_2_ is the difference in fugacity of CO_2_ between water and air, [*fCO*_2_ (water) – *fCO*_2_ (air)]. *k* was calculated according to the equation of Wanninkhof, (1992) [33] as given below:
(2)$$k=0.17u_{10}{\left(\frac{660}{\mathrm{Sc}}\right)}^{2/3}\;\mathrm{for}\;u_{10}\leq 3.6\;m/s$$
(3)$$k=\left(2.85u_{10}-9.65\right){\left(\frac{660}{Sc}\right)}^{0.5}\;\mathrm{for}\;3.6\leq u10\leq 13\;m/s$$
where *u*_10_ is the wind speed (m/s), and the Schmidt number (*Sc*) for CO_2_ was evaluated as per the formula:
(4)$$Sc=A-Bt+Ct^{2}-Dt^{3}$$
where *A* = 1992.1, *B* = 121.86, *C* = 3.54, *D* = 0.04227, and *t* = Temperature (^o^C) of water [34].

Ostwald solubility coefficient was calculated by the equation:ln β = A_1_ + A_2_(100/T)+ A_3_ In(T/100) +S[B_1_ + B_2_(T/100)+ B_3_(T/100)^2^]
where β = Ostwald solubility coefficient, ln = natural logarithm, i.e., log_e,_ and the T = water surface temperature (in degree Kelvin), and S = salinity in ppt.

A_1_ = −60.2409, A_2_ = 93.4517, A_3_ = 23.3585, B_1_ = 0.023517, B_2_ = −0.023656, B_3_ = 0.004703.

The positive magnitude of FCO_2_ interprets flux from water to air, and the negative value represents the reverse flux.

### 2.4. Statistical Analysis

The individual, as well as cumulative, datasets of biogeochemical properties were subjected to perform correlation and stepwise regression analysis, taking CO_2_ flux as the dependent variable and other associated parameters (WT, pH, salinity, chl *a*, F_0_, F_M_, PSII fluorescence, PBS fluorescence, PSII: PBS, yield, DO, and nutrients) as independent variables using SAAS 9.2 online version. Pearson correlation analysis of different physicochemical and biological parameters was carried out using SPSS Version 20.

## 3. Results

### 3.1. Biogeochemical Properties of Estuaries

#### 3.1.1. Biological Properties of Water in Estuaries

Our studies showed that the surface chl *a* ranged between 1.7 and 8.84, and 2.4 and 10.27 mg m^−3^ in the ME and DE estuaries, respectively (Table 1). The upper reach was found to have relatively higher chl *a* than the lower reach region and was strongly associated with high nutrient concentrations. The F_0_ and F_M_ showed site-wise, station-wise, and season-wise variations such as that of chlorophyll. F_0_ and F_M_ values were the highest in the post-monsoon period in the ME and DE. The relationship between chl *a* and fluorescence values (F_0_ and F_M_) was significantly linear (r = 0.983 and r = 0.987). The yield was also proportionately high in the post-monsoon period compared to other seasons in both estuaries. However, the yield values ranged from 0.32 to 0.49, which is relatively low. Nevertheless, yield variation was also significantly correlated with chl *a* (r = 0.962). The PBS fluorescence measured at 570 nm excitation was the highest in the post-monsoon period in both the estuaries. A comparison of pre-monsoon data of both the estuaries showed that the PBS fluorescence of the DE water was relatively higher than the ME water during this period. While a greater seasonal variation of PBS fluorescence was seen in the ME, it was proportionately less in the DE, indicating that the DE experienced a standing population of cyanobacteria throughout the year. Still, cyanobacterial blooming peaked during the post-monsoon months. For example, compared to the pre-monsoon, the increase in PBS fluorescence during the monsoon was higher in the ME (50%) than in the DE (6%). The same trend was also observed during the post-monsoon period (the increase was 50% and 26% in the ME and DE, respectively).

Unlike PBS, the PSII fluorescence in the DE during the monsoon decreased compared to the pre-monsoon season, whereas the same was more or less unchanged in the ME. During the monsoon to post-monsoon months, a significant rise in PS II fluorescence occurred in both the estuaries, though the rising rate was higher in the ME (90%) than in the DE (78%). The corresponding high value of the PBS/PSII ratio during the monsoon period was also noted in both estuaries. This indicates that during the monsoon months, the water turbidity limited the growth of green algae. Still, cyanobacterial growth was favored over green algae due to their low light adaptation. Irrespective of the season, the PBS/PS II ratio was higher in the DE than in the ME.

#### 3.1.2. Physicochemical Properties of Water in Estuaries

The variation of WT was not significant between the estuaries. However, seasonal variation at both the estuaries was prominent. Low temperatures were observed during the post-monsoon at both the estuaries (Figure 2). However, the DE showed more decline in WT (23.1 ± 0.1 °C) in the post-monsoon period as compared to the ME (28.8 ± 0.01 °C). Surface water pH attained the highest average value in the pre-monsoon in both the ME (8.197 ± 0.015) and DE (8.059 ± 0.038). The salinity of Mahanadi and Dhamra was primarily regulated by river discharges and was lowest during the monsoon at both the estuaries (4.2 ± 0.6 PSU in ME and 10.8 ± 3.5 PSU in DE). The nitrate concentrations were between 1.977 and 15.653 and 6.454 and 19.134 µmol L^−1^ in the ME and DE, respectively, whereas phosphate varied from 0.324 to 8.392 and 0.284 to 1.889 µmol L^−1^, respectively (Table 2). On the other hand, nitrite concentrations ranged from 0.048 to 2.678 in the ME and below the detection limit to 1.650 µmol L^−1^ in the DE. Similarly, the variations in DO concentration were observed in a range between 4.0 and 6.9 mg L^−1^ in the ME and 4.0 to 7.0 mg L^−1^ in the DE and were more or less consistent with chl *a* (Table 3, Table 4 and Table 5; r ≥ 0.4; *p* = 0.05), suggesting biological processes have significant control over water biogeochemistry. The *f*CO_2_ (water) ranged between 240.5 and 1268.0 µatm in the ME and 208.8 and 485.6 µatm in the DE, whereas the variation of *f*CO_2_ (air) was 378.2–432.7 in the ME and 378.8–408.7 µatm in the DE. In the ME, the highest TA and DIC were observed in the pre-monsoon (1873.4 ± 213.1 µmol kg^−1^ and 1415.9 ± 49.0 µmol kg^−1^, respectively). In the DE, the highest TA was observed in the pre-monsoon (1873.4 ± 213.1 µmol kg^−1^), and the highest DIC was observed in the post-monsoon (1683.5 ± 28.8 µmol kg^−1^).

### 3.2. Air–Water CO_2_ Fluxes in Estuaries and Their Drivers

The regression models built from the whole dataset showed that the overall air–water CO_2_ fluxes (FCO_2_) were not significantly dependent on any biological parameter. However, the CO_2_ flux of the ME during the monsoon was significantly dependent on pH (Table 6). In contrast, in the DE, the CO_2_ flux depended on pH and salinity. The FCO_2_ in the ME, irrespective of the seasons, was significantly driven by biological parameters such as chl *a*, yield, and PBS: PSII, whereas only PBS: PSII significantly affects FCO_2_ in the DE among the biological parameters. The estuarine effect on biological parameters is more prominent, but the physicochemical parameters such as WT, pH, and salinity were the governing parameters in both the estuaries.

### 3.3. Correlation among Physicochemical and Biological Parameters of Estuaries

The PS II fluorescence intensity was directly correlated with the chl *a* content of the water sample (r≥0.99). However, in the monsoon season, it was found less due to the water turbidity and greater PBS fluorescence. There was an inter-seasonal variation of PBS fluorescence at both the study sites, and such variations were found to be significant. While in the ME, no significant improvement in the PSII fluorescence was observed in the monsoon compared to the pre-monsoon, the corresponding PBS fluorescence increment was relatively high. However, increased phytoplankton density in the post-monsoon, presumably driven by nutrient availability, led to higher PBS and PSII fluorescence compared to the monsoon season. On the other hand, there was a significant reduction of the PSII fluorescence, and no significant change in the PBS fluorescence was noted in the monsoon than in the pre-monsoon of the DE. However, the trend from monsoon to post-monsoon in the DE was similar to the ME. The fluorescence parameters were found to be significantly correlated with water temperature, thus showing that the phytoplankton density/diversity of the estuaries was well-influenced by water temperature (r = −0.40 to −0.46) but independent of the pH changes of the surface water (r = −0.002 to −0.114).

## 4. Discussion

### 4.1. Biological Bloom and Behavior in Estuaries

To the best of our knowledge, this study reports for the first time the efficiency of Chl *a* fluorescence as a marker for predicting the air–water CO_2_ flux in estuaries. The photosynthetic activity of phytoplankton in the coastal ocean plays a critical role in maintaining the gradient of CO_2_ concentration between the surface layer of the ocean and its overlying atmosphere, mainly regulating the air–water CO_2_ exchange and its spatial distribution. Photosynthesis in the coastal ocean reduces the CO_2_ concentration by converting dissolved inorganic carbon into an organic form of carbon that decomposes and remineralizes back to dissolved organic carbon at ocean depth. This biological cycle is responsible for the net movement of carbon from the surface layer to the subsurface and subsequently to the ocean depth, referred to as the “biological pump”. This study found that the biological pump was very active during the post-monsoon in both the estuaries. The chl *a* and the associated photosynthetic performance parameters were high in the post-monsoon compared to other seasons. Irrespective of seasons, the biological parameters were higher in magnitude in the DE compared to the ME. While PS II fluorescence was not much influenced by the change of season from the pre-monsoon to monsoon in the ME, a significant decrease of fluorescence intensity was observed in the DE during the period indicating that monsoon-induced increase in turbidity of DE water limited the growth of green phytoplankton and, consequently, directed the resources for an increase in density of cyanobacteria as the latter are adapted to low-light intensities. The corresponding rise in PBS fluorescence corroborated this. Throughout the year, the photosynthetic yield of the DE was high compared to that of the ME, indicating a high planktonic producer load in the DE. No significant seasonal change in yield was observed in the DE, but in the ME, the yield significantly increased in the post-monsoon season as compared to the monsoon months, and it was attributed to the high growth of green phytoplankton. When the biological data of both the estuaries were pooled together, the correlation analysis showed that CO_2_ flux was negatively correlated with the fluorescence parameters. Thus, it can be inferred that the growth and photosynthetic performance of phytoplankton, irrespective of the qualitative variation, have a significant impact on the CO_2_ flux of the estuaries.

It has also been reported that more than 1 million tons of carbon are fixed into organic matter each day by ocean biological fixation [35]. Due to the biological control of the dissolved inorganic carbon in the ocean, the current average atmospheric CO_2_ concentration of earth is ~410 ppm, which would be 150–200 µatm greater than the current value in the absence of ocean biological fixation. From various research, it is also clear that the coastal bloom takes up a large portion of anthropogenic CO_2_, adding organic carbon to the coastal ocean and contributing to the inner ocean’s primary production [16,36,37].

Chl *a* fluorescence has become an essential analytical parameter in oceanography and limnology to predict phytoplankton productivity and examine the diversity of producers in the aquatic systems [38,39]. Active chl *a* fluorescence measurement evaluates the efficiency by which absorbed light is utilized by photosynthesis. The fluorescence ratios have also become useful to determine the dominance of one group of algae or the other in a system. Because of the efficient transfer of energy to reaction center chlorophylls, the fluorescence emission from chl *a* usually acts as an indicator of the phytoplankton density vis-a-vis primary production [40]. Fluorescence spectra are useful for determining the phytoplankton composition and for measuring the dynamics of change in phytoplankton dominance. However, samples’ excitation and emission characteristics vary with the qualitative and quantitative changes in phytoplankton composition [41]. Nevertheless, PS II and PBS’s intensity are beneficial for predicting phytoplankton dominance and production potential [42]. The relationships between algal growth and environmental factors have been previously discussed in many Indian estuaries, but the use of the physiological parameters of algae about CO_2_ sequestration is rarely discussed [43].

### 4.2. Temporal and Spatial Relationship of Physicochemical Properties of Estuaries

The Bay of Bengal receives a good amount of freshwater discharge during the monsoon season; therefore, a weakened coastal upwelling was observed. During the period of our observations, the DE was found to be more saline than ME (Figure 3 and Figure 4) because the Mahanadi river has a greater number of tributaries, and it receives more fresh water in comparison to Dhamra (confluence of Brahmani and Baitarani) [44]. Seasonal variations in chl *a* and nutrients were primarily due to river discharge and other associated biogeochemical processes. The source of nutrients is usually associated with low salinity, and temperature signifies that the nutrient sources are river-driven. During monsoon, the chl a concentration was low despite the high amount of nutrients. This led to the conclusion that even though there were sufficient nutrients by land contribution during the monsoon season, the growth of chlorophyll-dominated phytoplankton was limited by suspended particulate matters, and cyanobacteria well exploited the available resources because of their low light adaptability. In the Arabian Sea, the monsoon-driven changes in water chemistry cause a shift in the dominance of phytoplankton, the cyanobacteria being the dominant group during monsoon months [45]. The strong seasonal variation of PBS/PS II ratio and its high values in the monsoon seasons in this study supports the conclusion that the cyanobacteria are abundant during the monsoon relative to green algae and diatoms in post-monsoon conditions. It is essential to mention that the increase in chl *a* and associated changes in nutrients, WT, and salinity can be useful to evaluate possible mechanisms regulating the CO_2_ flux.

### 4.3. Drivers of Variations of Air–Water CO_2_ Exchanges and Their Prediction

The detailed estimation of air–water CO_2_ fluxes in our study revealed sharp seasonal changes in the Mahanadi and Dhamra estuaries (Figure 5). The biological parameters had significantly driven the fluxes in the ME. However, in the DE, the physicochemical variables such as WT, pH, and salinity regulated the flux behavior rather than the biological parameters. Chl *a* reveal a negative correlation with CO_2_ flux, suggesting that phytoplankton growth has a vital role in controlling the capacity of the estuary to atmospheric uptake CO_2_. The stepwise regression model revealed that FCO_2_ could be predicted precisely with the help of specific physicochemical and biological parameters. This could be a valuable tool for understanding the regulation mechanism of these parameters over CO_2_ flux and vice-versa.

## 5. Conclusions

The Mahanadi estuary, Bay of Bengal, India, acted as a source of CO_2_ fluxes in the monsoon and post-monsoon. On the contrary, the Dhamra estuary acted as a sink of CO_2_ fluxes during the monsoon and a weak source in the post-monsoon. There is not much variation in the sink strength in the pre-monsoon between the ME and DE. The upper reaches (near the sea) of the estuaries were characterized by more saline water, with more pH and sink strength. Air–water CO_2_ flux and surface chl *a* concentration were negatively correlated, suggesting that photosynthesis is one of the primary processes responsible for the substantial CO_2_ sequestration in estuaries and other coastal oceans. The biological CO_2_ drawdown is site-specific and significant on a large area over a long temporal scale. Still, the seasonal cycle of CO_2_ flux is regulated less by biological and more by physicochemical parameters such as WT, pH, and salinity. The algal composition and assemblage showed distinct characteristics due to variable nutrient concentration and water transparency. Our stepwise regression model suggested that biological parameters are more site-specific and sensitive to WT, pH, and salinity. Long-term analysis of biological parameters with real-time air–water CO_2_ exchanges should be monitored over a spatial scale to judge the climate change feedback behavior of estuaries in the Bay of Bengal. However, the collected data in this study and other studies may help understand the future climate trend. Further, it may help the global community understand the flux behavior and associated dynamics when seasonal data from such marginally studied estuaries are considered for scale-up processes.

## Figures and Tables

**Figure 1 life-12-01536-f001:**
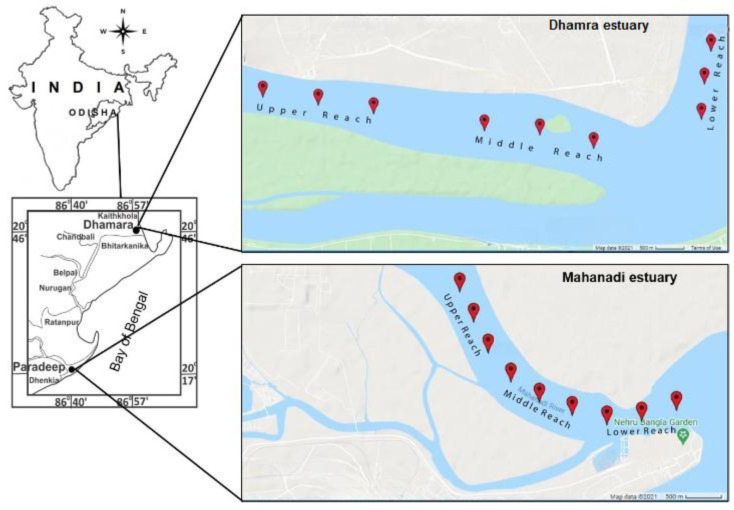
Sketch map of two important estuaries of northeastern Bay of Bengal showing three distinct reaches of Mahanadi estuary and Dhamra estuary.

**Figure 2 life-12-01536-f002:**
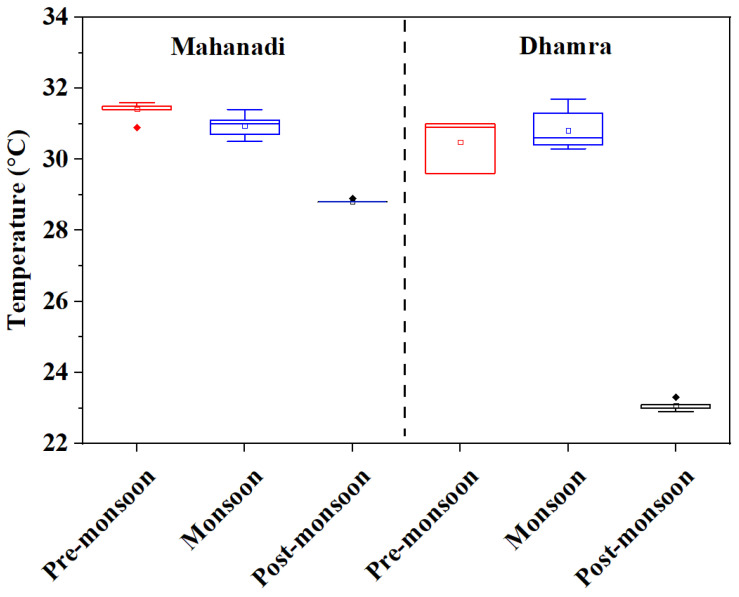
Seasonal variability of temperature (*p* < 0.001) in Mahanadi and Dhamra estuaries.

**Figure 3 life-12-01536-f003:**
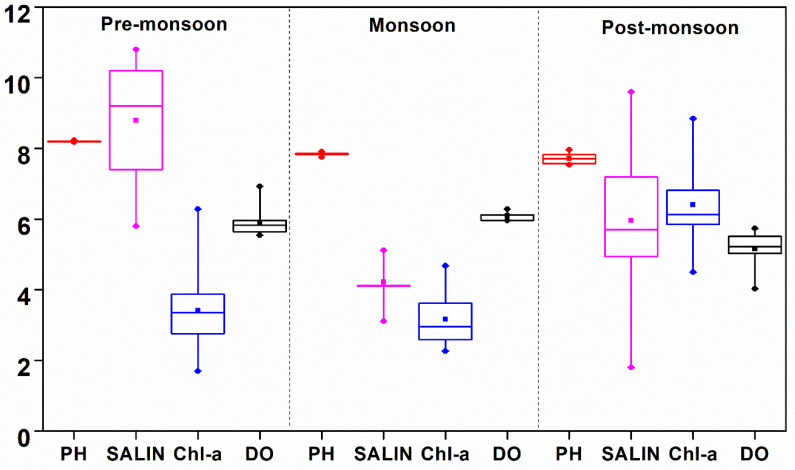
Seasonal variability of pH, salinity, chl *a,* and dissolved oxygen in Mahanadi estuary (*p* < 0.001).

**Figure 4 life-12-01536-f004:**
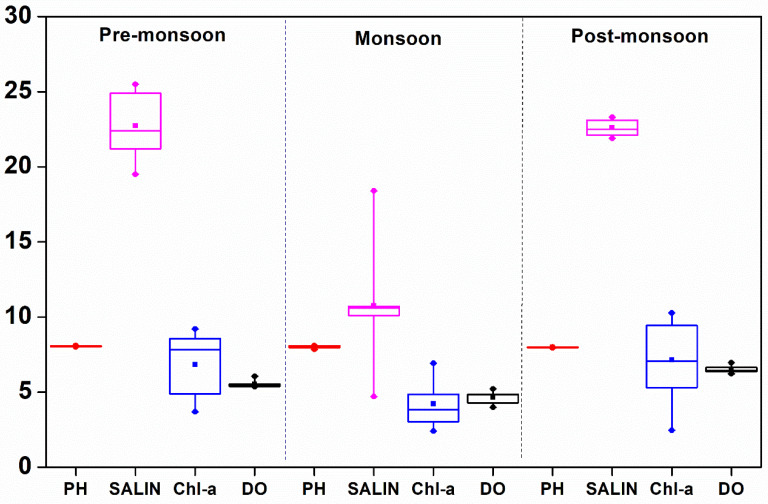
Seasonal variability of pH (*p* < 0.05), salinity (*p* < 0.001), chl *a* (*p* < 0.05), and dissolved oxygen (*p* < 0.001) in Dhamra estuary.

**Figure 5 life-12-01536-f005:**
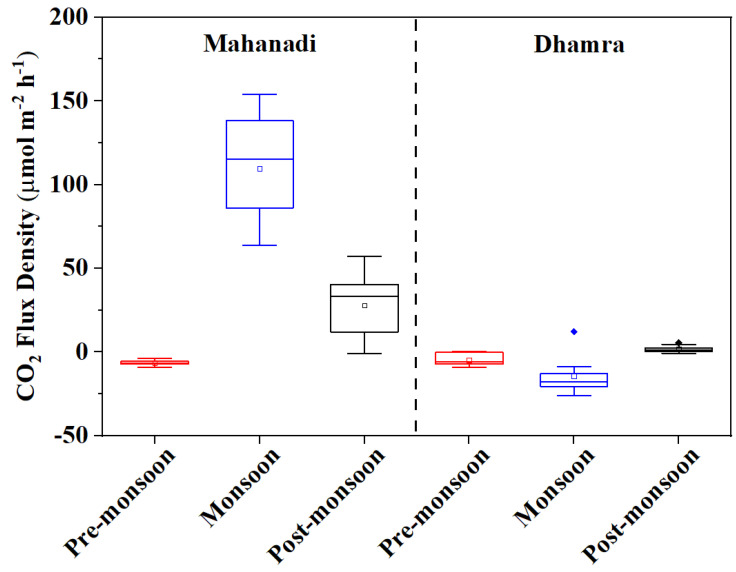
Seasonal variability of CO_2_ fluxes (*p* < 0.001) in Mahanadi and Dhamra estuaries.

**Table 1 life-12-01536-t001:** Seasonal mean ± standard deviation (SD) along with a range of essential biological parameters in the Mahanadi and Dhamra estuaries.

Parameters		Mahanadi			Dhamra		
		Pre-Monsoon	Monsoon	Post-Monsoon	Pre-Monsoon	Monsoon	Post-Monsoon
Chl *a* (mg m^−3^)	Mean	3.41 ± 1.36 1.71–6.28	3.16 ± 0.77 2.26–4.68	6.40 ± 1.23 4.51–8.84	6.83 ± 2.13 3.68–9.20	4.23 ± 1.69 2.40–6.92	7.15 ± 2.70 2.46–10.27
	Range						
Fluorescence minimum	Mean	44.73 ± 8.01 34.27–59.48	41.10 ± 7.36 34.24–56.34	67.86 ± 9.18 52.45–82.67	69.21 ± 15.64 45.78–87.95	50.66 ± 13.93 34.53–71.69	72.41 ± 18.95 37.23–92.45
	Range						
Fluorescence maximum	Mean	70.88 ± 16.5 51.78–103.86	64.86 ± 15.12 50.35–97.44	119.34 ± 20.07 88.67–153.75	125.8 ± 34.57 73.95–171.83	84.82 ± 29.91 51.68–130.79	134.08 ± 45.68 53.24–184.62
	Range						
Yield	Mean	0.36 ± 0.03 0.32–0.43	0.36 ± 0.03 0.32–0.42	0.43 ± 0.02 0.40–0.46	0.44 ± 0.03 0.38–0.49	0.39 ± 0.05 0.33–0.45	0.44 ± 0.07 0.30–0.51
	Range						
Phycobilisome fluorescence	Mean	17.44 ± 6.93 8.72–32.17	26.42 ± 6.25 18.34–38.94	39.27 ± 7.83 27.61–54.77	35.7 ± 10.94 19.32–48.21	37.37 ± 14.28 22.35–60.54	46.07 ± 17.42 16.04–66.27
	Range						
PSII fluorescence	Mean	3.45 ± 1.38 1.71–6.42	3.47 ± 0.79 2.53–4.98	6.60 ± 1.28 4.62–9.17	6.49 ± 1.98 3.53–8.65	3.94 ± 1.49 2.43–6.31	6.98 ± 2.52 2.54–9.94
	Range						
PBS/PSII	Mean	5.07 ± 0.11 4.89–5.20	7.61 ± 0.25 7.24–7.96	5.94 ± 0.08 5.74–6.03	5.50 ± 0.10 5.34–5.64	9.51 ± 0.30 8.88–9.83	6.55 ± 0.21 6.31–6.97
	Range						

**Table 2 life-12-01536-t002:** Seasonal mean ± SD, along with a range of essential physicochemical parameters in the Mahanadi and Dhamra estuaries.

Parameters		Mahanadi			Dhamra		
		Pre-Monsoon	Monsoon	Post-Monsoon	Pre-Monsoon	Monsoon	Post-Monsoon
CO_2_ flux (µmol m^−2^ h^−1^)	Mean	−6.5 ± 1.6 −9.2–−3.5	109.5 ± 31.5 63.8–154.1	28 ± 20.1 −0.7–57.1	−4.7 ± 3.7 −9.2–0.3	−14.2 ± 11.1 −25.8–12.2	1.8 ± 2.2 −0.7–5.6
	Range						
*f*CO_2_ (water) (µatm)	Mean	270.3 ± 17.4 240.5–289.6	759.4 ± 109.9 600.3–910.9	840.0 ± 294.2 417.6–1268.0	300.2 ± 31.4 270.5–365.2	258.8 ± 38.5 208.8–327.3	432.2 ± 31.2 395.4–485.6
	Range						
*f*CO_2_ (air) (µatm)	Mean	381.7 ± 2.724 379.4–388.2	379.7 ± 2.681 378.2–386.8	421.7 ± 7.397 414.1–432.7	382.2 ± 2.118 378.8–384.9	386.6 ± 3.9 382.2–394.90	406.3 ± 0.993 405.4–408.7
	Range						
TA (µmol kg^−1^)	Mean	1564.0 ± 54.63 1476.9–1669.5	1240.8 ± 105.8 1123.8–1457.9	1284.4 ± 250.2 1012.4–1669.3	1873.4 ± 213.1 1513.0–2151	1022.0 ± 52.3 919.8–1079.7	1831.9 ± 29.1 1769.2–1876.2
	Range						
DIC (µmol kg^−1^)	Mean	1415.9 ± 49.0 1346.8–1502.2	1234.6 ± 105.6 1112.5–1448.9	1270.1 ± 249.5 998.1–1648.7	1606.4 ± 140.9 1329.3–1774.6	934.2 ± 46.4 853.5–998.0	1683.5 ± 28.8 1622.3–1731.1
	Range						
pH	Mean	8.197 ± 0.015 8.180–8.230	7.847 ± 0.048 7.750–7.911	7.716 ± 0.153 7.541–7.962	8.059 ± 0.038 8.011–8.101	8.02 ± 0.088 7.840–8.101	7.974 ± 0.021 7.940–8.001
	Range						
Water temperature (°C)	Mean	31.4 ± 0.21 30.9–31.6	30.9 ± 0.32 30.5–31.4	28.8 ± 0.01 28.8–28.9	30.5 ± 0.71 29.6–31.0	30.8 ± 0.62 30.3–31.7	23.1 ± 0.13 22.9–23.3
	Range						
Salinity (psu)	Mean	8.8 ± 1.8 5.8–10.8	4.2 ± 0.6 3.1–5.1	6 ± 2.3 1.8–9.6	22.7 ± 2.1 19.5–25.5	10.8 ± 3.5 4.7–18.4	22.6 ± 0.5 21.9–23.3
	Range						
D.O. (mg L^−1^)	Mean	5.9 ± 0.4 5.5–6.9	6.1 ± 0.1 6.0–6.3	5.2 ± 0.5 4.0–5.7	5.5 ± 0.2 5.3–6.1	4.6 ± 0.4 4.0–5.2	6.5 ± 0.2 6.2–7.0
	Range						
Nitrite (µmol L^−1^)	Mean	1.562 ± 0.661 0.52–2.678	0.262 ± 0.035 0.209–0.313	0.100 ± 0.038 0.048–0.152	0.737 ± 0.572 0.052–1.650	0.309 ± 0.088 0.174–0.483	0.247 ± 0.25 BDL–0.774
	Range						
Nitrate (µmol L^−1^)	Mean	7.839 ± 4.771 2.639–15.038	14.731 ± 0.752 13.542–15.653	4.259 ± 2.579 1.977–10.728	9.028 ± 1.387 7.358–10.915	8.114 ± 1.81 6.454–12.202	16.391 ± 1.621 14.235–19.134
	Range						
Ammonia (µmol L^−1^)	Mean	0.055 ± 0.108 BDL–0.338	1.217 ± 0.36 0.832–1.756	1.351 ± 1.658 BDL–5.273	7.976 ± 11.246 BDL–34.315	1.127 ± 0.973 0.211–3.464	5.369 ± 6.809 0.658–20.516
	Range						
Phosphate (µmol L^−1^)	Mean	2.554 ± 1.198 1.348–5.037	5.819 ± 1.423 4.250–8.392	0.967 ± 0.504 0.324–1.758	0.467 ± 0.124 0.284–0.659	1.485 ± 0.194 1.289–1.889	0.363 ± 0.203 0.137–0.811
	Range						

**Table 3 life-12-01536-t003:** Pearson correlation analysis of different physicochemical and biological parameters using the cumulative dataset of Mahanadi and Dhamra estuaries.

	FCO_2_ (Flux Density)	PH	Water Temperature	Salinity	Chl *a*	F_0_	F_m_	Yield	PBS	PS II	PBS/PSII	DO	NO_2_^−^	NO_3_^−^	NH_3_	PO_4_^3−^
FCO_2_ (Flux density)	1															
PH	**−0.597 ****	1														
Water temperature	0.152	0.186	1													
salinity	**−0.504 ****	0.313 *	**−0.535 ****	1												
Chl *a*	−0.327*	−0.016	**−0.448 ****	**0.438 ****	1											
F_0_	**−0.359 ****	−0.017	**−0.459 ****	**0.441 ****	**0.990 ****	1										
F_m_	**−0.351 ****	−0.002	**−0.468 ****	**0.454 ****	**0.994 ****	**0.996 ****	1									
Yield	−0.331 *	−0.024	**−0.401 ****	**0.432 ****	**0.962 ****	**0.9 57****	**0.965 ****	1								
PBS	−0.222	−0.114	**−0.456 ****	**0.289 ***	**0.902 ****	**0.896 ****	**0.906 ****	**0.890 ****	1							
PS II	−0.262	−0.076	**−0.457 ****	**0.381 ****	**0.994 ****	**0.984 ****	**0.985 ****	**0.949 ****	**0.892 ****	1						
PBS/PSII	0.133	−0.160	0.077	−0.228	−0.206	−0.216	−0.198	−0.159	0.196	−0.238	1					
DO	0.265	0.056	**−0.452 ****	0.271 *	0.043	0.029	0.049	0.013	−0.077	0.072	**−0.394 ****	1				
NO_2_^−^	−0.136	0.217	0.114	0.207	−0.087	−0.109	−0.096	0.008	−0.196	−0.105	−0.229	−0.017	1			
NO_3_^−^	0.339 *	0.008	**−0.443 ****	0.292 *	−0.016	−0.057	−0.022	−0.037	0.074	−0.011	0.156	**0.628 ****	−0.100	1		
NH_3_	−0.123	0.086	−0.109	**0.395 ****	**0.360 ****	**0.361 ****	**0.365 ****	0.334 *	0.244	0.328 *	−0.122	0.070	−0.102	0.036	1	
PO_4_^3−^	**0.718 ****	−0.104	**0.432 ****	**−0.612 ****	**−0.571 ****	**−0.587 ****	**−0.570 ****	**−0.566 ****	**−0.428 ****	**−0.534 ****	0.219	0.157	−0.096	0.257	−0.218	1

* Correlation is significant at the 0.05 level. ** Correlation is significant at the 0.01 level.

**Table 4 life-12-01536-t004:** Pearson correlation analysis of different physicochemical and biological parameters using the dataset of Mahanadi estuary.

	FCO_2_ (Flux Density)	PH	Water Temperature	Salinity	Chl *a*	F_0_	F_m_	Yield	PBS	PS II	PBS/PSII	DO	NO_2_^−^	NO_3_^−^	NH_3_	PO_4_^3−^
FCO_2_ (Flux density)	1															
PH	**−0.716 ****	1														
Water temperature	**−0.601 ****	0.479 *	1													
salinity	**0.532 ****	−0.001	−0.485 *	1												
Chl *a*	0.105	0.283	−0.288	0.435 *	1											
F_0_	0.089	0.286	−0.302	0.436 *	**0.993 ****	1										
F_m_	0.098	0.273	−0.309	0.427 *	**0.996 ****	**0.997 ****	1									
Yield	0.070	0.297	−0.223	0.407 *	**0.954 ****	**0.951 ****	**0.960 ****	1								
PBS	−0.149	0.223	−0.283	0.063	**0.867 ****	**0.871 ****	**0.881 ****	**0.855 ****	1							
PS II	0.145	0.247	−0.334	0.473 *	**0.997 ****	**0.991 ****	**0.994 ****	**0.945 ****	**0.853 ****	1						
PBS/PSII	**−0.544 ****	−0.099	0.288	**−0.918 ****	−0.462 *	−0.457 *	−0.442 *	−0.411 *	0.000	−0.492 **	1					
DO	**0.608 ****	−0.250	**−0.829 ****	**0.781 ****	0.460*	0.464 *	0.464 *	0.378	0.240	**0.509 ****	**−0.644 ****	1				
NO_2_^−^	0.117	0.027	0.116	0.221	−0.091	−0.121	−0.106	0.036	−0.210	−0.096	−0.226	−0.056	1			
NO_3_^−^	**0.625 ****	**−0.541 ****	**−0.903 ****	**0.502 ****	0.299	0.319	0.315	0.213	0.251	0.351	−0.343	**0.841 ****	−0.193	1		
NH_3_	0.077	0.245	0.032	0.298	0.324	0.331	0.320	0.284	0.127	0.308	−0.329	0.148	−0.121	−0.073	1	
PO_4_^3−^	**−0.589 ****	−0.005	**0.580 ****	**−0.886 ****	**−0.505 ****	**−0.499 ****	**−0.498 ****	−0.450 *	−0.156	**−0.536 ****	**0.884 ****	**−0.796 ****	−0.198	**−0.512 ****	−0.251	1

* Correlation is significant at the 0.05 level. ** Correlation is significant at the 0.01 level.

**Table 5 life-12-01536-t005:** Pearson correlation analysis of different physicochemical and biological parameters using the dataset of Dhamra estuary.

	FCO_2_ (Flux Density)	PH	Water Temperature	Salinity	Chl *a*	F_0_	F_m_	Yield	PBS	PS II	PBS/PSII	DO	NO_2_^−^	NO_3_^−^	NH_3_	PO_4_^3−^
FCO_2_ (Flux density)	1															
PH	**−0.543 ****	1														
Water temperature	0.066	**0.734 ****	1													
salinity	**−0.577 ****	0.390*	0.197	1												
Chl *a*	−0.294	−0.353	**−0.740 ****	−0.318	1											
F_0_	−0.354	−0.338	**−0.760 ****	−0.232	**0.983 ****	1										
F_m_	−0.343	−0.340	**−0.762 ****	−0.263	**0.987 ****	**0.998 ****	1									
Yield	−0.285	−0.373	**−0.702 ****	−0.300	**0.957 ****	**0.950 ****	**0.958 ****	1								
PBS	0.053	**−0.545 ****	**−0.733 ****	**−0.587 ****	**0.919 ****	**0.891 ****	**0.903 ****	**0.898 ****	1							
PS II	−0.241	−0.384*	**−0.742 ****	−0.366	**0.998 ****	**0.976 ****	**0.981 ****	**0.952 ****	**0.940 ****	1						
PBS/PSII	**0.907 ****	**−0.506 ****	0.005	**−0.721 ****	−0.183	−0.235	−0.211	−0.140	0.208	−0.128	1					
DO	0.290	0.280	**0.683 ****	0.021	**−0.650 ****	**−0.669 ****	**−0.666 ****	**−0.569 ****	**−0.563 ****	**−0.644 ****	0.250	1				
NO_2_^−^	**−0.526 ****	0.801 **	**0.624 ****	0.498 **	−0.344	−0.349	−0.351	−0.321	**−0.582 ****	−0.378	**−0.611 ****	0.255	1			
NO_3_^−^	**0.646 ****	0.036	**0.528 ****	−0.326	**−0.545 ****	**−0.632 ****	**−0.606 ****	**−0.502 ****	−0.302	**−0.510 ****	**0.625 ****	**0.574 ****	−0.040	1		
NH_3_	0.328	**−0.523 ****	−0.375	−0.319	0.219	0.241	0.250	0.248	0.361	0.244	0.366	−0.135	−0.470 *	0.038	1	
PO_4_^3−^	**0.637 ****	0.086	**0.505 ****	−0.317	**−0.651 ****	**−0.673 ****	**−0.652 ****	**−0.643 ****	−0.369	**−0.614 ****	**0.694 ****	**0.490 ****	−0.085	**0.727 ****	0.058	1

* Correlation is significant at the 0.05 level. ** Correlation is significant at the 0.01 level.

**Table 6 life-12-01536-t006:** Regression models of FCO_2_ with related physicochemical and biological parameters.

	Regression Equation	r^2^ (*p* ≤ 0.05)
Mahanadi		
Pre-Monsoon	FCO_2_ = −235.2 + 30.5 pH − 1.4 salinity − 0.12 Fm + 5.3NH_4_	r^2^ = 0.94
Monsoon	FCO_2_ = 4436.2 − 552.4 pH	r^2^ = 0.69
Post-Monsoon	FCO_2_ = 9397.4 − 171.1 pH − 279.8 Water temperature +179.1 NO_2_	r^2^ = 0.92
Dhamra		
Pre-Monsoon	FCO_2_ = 160.3 − 5.41 temperature	r^2^ = 0.95
Monsoon	FCO_2_ = 867.7 − 108.7 pH − 0.92 salinity	r^2^ = 0.93
Post-Monsoon	FCO_2_ = 363.7 − 58.6 pH + 5.4 Water temperature 7 − 0.91 salinity	r^2^ = 0.98
Mahanadi Annual	FCO_2_ = 620.4 − 160.5 pH +24.9 temperature +17.5 Chl *a* − 836.8 yield + 29.3 PBS:PSII	r^2^ = 0.94
Dhamra Annual	FCO_2_ = 965.9 − 116.1 pH − 0.39 salinity − 4.6 PBS:PSII	r^2^ = 0.90
Mahanadi and Dhamra Annual	FCO_2_ = 1079.8 − 162.6 pH + 4.0 temperature + 13.8 DO +1.9 nitrate +10.1 phosphate	r^2^ = 0.86

## Data Availability

The authors themselves have measured the primary data used in this study and can be provided by the corresponding author on reasonable request and the secondary data used in this study is duly cited all through the manuscript.

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
