# Peer review of "The Interaction of Seasons and Biogeochemical Properties of Water Regulate the Air–Water CO2 Exchanges in Two Major Tropical Estuaries, Bay of Bengal (India)"

_life, 2022, doi:10.3390/life12101536_

Round 1
Reviewer 1 Report
The paper deals with a study to evaluate how the seasonal variability of biogeochemical properties of water affects air-sea CO2 exchanges in two specific estuaries. The topic is worthy of interest and the used methods are clearly indicated.
In my opinion the section "Introduction" needs to be changed before publication, including a broader state of the art and moving lines 58-92 to Section 2.1 as those lines provide a description of the estuaries instead of an introduction on the subject.
I appreciate the statistical analysis made by the Authors, there is no quantification of the samples used by them. How many samples do the Authors use in each season? How many years of data do the Authors analyzed? This is critical to understand the relevance of the study.
Are the samples used in the study representative of the common characteristics of the territory? Please discuss this more accurately.
Long-term measurement is a prerogative for climate change analysis, as the Authors reported in lines 390-391, can this study be considered valid for observing a climate trend? Please add a discussion on this.
Minor remarks:
-line 65: delete the space between "regions" and "of Eastern"
-line 72: add a space before 13,590 km2
-line 126: acronym chl-a is not defined
-line 195: delete the full stop before Results
-line 213: change "year" with "years"
-line 217: the acronym SD is not defined
-line 237: delete the space between "6.454" and "to"
-line 241: delete the space between "mg L-1" and "in" and delete the space between "and" and "4.0"
-line 329: Eggert et al., 2006 is missing in the Reference list, please add it and modify the order of the reference, accordingly.
-line 379: please add a space between "sea)" and "of"
Author Response
Author’s response to Reviewer #1
Respected reviewer,
Thank you very much for your insightful and constructive comments on our manuscript. We have responded all of your comments and addressed the raised concerns as much as possible for the manuscript entitled “The interaction of seasons and biogeochemical properties of water regulate the air-water CO2 exchanges in two major tropical estuaries, Bay of Bengal (India)” (Manuscript ID: life-1908774). We have edited the manuscript and corrected it to the best of our ability. Your comments and our responses to each comment are stated below. We believe that the revised manuscript will be worthy of being accepted for publication in the Special Issue of Life.
Comment 1: In my opinion the section "Introduction" needs to be changed before publication, including a broader state of the art and moving lines 58-92 to Section 2.1 as those lines provide a description of the estuaries instead of an introduction on the subject.
Response 1: We take the opportunity to thank you for your comments and suggestions that helped us a lot to remove several ambiguities from the manuscript and enhance its scientific quality. We have added two paragraphs to the Introduction Section according to your suggestion (L37-L53) to improve its scientific thrust and we have deleted the sentences to enhance clarity.
Comment 2: I appreciate the statistical analysis made by the Authors, there is no quantification of the samples used by them. How many samples do the Authors use in each season? How many years of data do the Authors analyzed? This is critical to understand the relevance of the study.
Response 2: First of all, thank you for appreciating our effort in the statistical analysis. We have mentioned quantification of samples between L179-L182. We understand the fact that the sampling time and frequency are critical to understanding the characteristics of any estuarine system. Keeping the above in view, we have collected the samples in duplicates from 18 stations (9 stations of Dharma Estuary and 9 stations of Mahanadi Estuary) in each season for a year.
Comment 3: Are the samples used in the study representative of the common characteristics of the territory? Please discuss this more accurately.
Response 3: We have chosen the estuaries for the study differ from each other in so many aspects such as discharge, salinity, turbidity, and the density and diversity of phytoplankton, etc. which is well evidenced in the present results and other publications of Pattanaik et al. (2019) and (2020). However, the samples collected from an estuary always share common characteristics which are also observed in this study.
Comment 4: Long-term measurement is a prerogative for climate change analysis, as the Authors reported in lines 390-391, can this study be considered valid for observing a climate trend? Please add a discussion on this.
Response 4: The climate change study needs at least 30 years of data for drawing some valid conclusions. This study holds good to observe the seasonality and can not be considered for climate trends as such. However, it may help the global community in understanding the flux behavior and associated dynamics when seasonal data from such marginally studied estuaries are taken into consideration for scale-up processes. Further, the collected data in this study, and other studies may cumulatively help to understand the climate trend in the future.
*Minor remarks by the reviewer have been attended properly.

Reviewer 2 Report
1. I think the key words need to highlight the bay of Bengal, and the current samples are too few to be representative.
2. The introduction is inconsistent with the content of the following article, and needs to be rewritten. Please add the content on the experimental principle and delete the irrelevant content
3. In line 199-200: The upper reach was found to have relatively higher chl a than the lower reach region and was strongly associated with high nutrient concentrations. However, we can't get this information from Table 1. And the meanwhile, in the line 206: the yield values ranged from 0.32 to 0.49 which is relatively low. This data is different from the data listed in the table 1.
4. Please explain what the different colors in the Fig. 2 represent.
5. In the line 329-330: Fluorescence spectra are useful for determining the phytoplankton composition and for measuring the dynamics of change in phytoplankton dominance.
6. Cyanobacteria, green algae, and diatoms are mentioned in the manuscript, but it does not explain how to determine the abundance through fluorescence spectra. More explanation is needed here
7. In the line 287-288: To the best of our knowledge, this study reports for the first time the efficiency of Chl a fluorescence as a marker for predicting the air-water CO2 flux in estuaries. The authors only studied 18 stations in the Mahanadi and Dhamra estuaries in the Bay of Bengal, the amount of data is too small and not representative of all estuaries.
8. Please unify the abbreviations in the manuscript.
9. Please add p value in all pictures. Moreover, the form of the picture is too simple.
Author Response
Author’s response to Reviewer #2
Respected reviewer,
Thank you very much for your insightful and constructive comments on our manuscript. We have carefully gone through all your comments on our manuscript entitled “The interaction of seasons and biogeochemical properties of water regulate the air-water CO2 exchanges in two major tropical estuaries, Bay of Bengal (India)” (Manuscript ID: life-1908774) and we have modified and edited the manuscript to the best of our capabilities. The respective responses to all your comments are listed below. We hope the revised manuscript is worth accepting for publication in the Special Issue of Life.
Comment 1: I think the keywords need to highlight the Bay of Bengal, and the current samples are too few to be representative.
Response 1: The idea to highlight the Bay of Bengal in keywords is great. The two estuaries are the representatives of the Bay of Bengal and Mahanadi estuary is one of the big estuaries of the Bay of Bengal. Therefore, we have added Bay of Bengal to the keywords. In the Bay of Bengal, there are many big and small estuaries. We want to present the behavior of two major estuaries ending in the Bay of Bengal and not the behavior of the entire series of estuaries of the Bay of Bengal. Therefore, 9 samples per estuary and, 18 samples for two estuaries representing the three reaches may be adequate in our understanding.
Comment 2: The introduction is inconsistent with the content of the following article, and needs to be rewritten. Please add the content on the experimental principle and delete the irrelevant content.
Response 2: We have revised the Introduction section according to your suggestions.
Comment 3: In lines 199-200: The upper reach was found to have relatively higher chl a than the lower reach region and was strongly associated with high nutrient concentrations. However, we can't get this information from Table 1. And meanwhile, in line 206: the yield values ranged from 0.32 to 0.49 which is relatively low. This data is different from the data listed in table 1.
Response 3: We prepared the table content as per the seasonal variation however lines 258-260 described the spatial variation of chl a (data not shown). The yield value ranged from 0.32 to 0.51 as a whole (0.49 is a typo) which was real-time data and is relatively low.
Comment 4: Please explain what the different colors in the Fig. 2 represent.
Response 4: The different colors are used for three different seasons to make the figure more distinct.
Comment 5: In the line 329-330: Fluorescence spectra are useful for determining the phytoplankton composition and for measuring the dynamics of change in phytoplankton dominance.
Response 5: We have measured the fluorescence change of the samples with 570 nm and 440 nm excitation. The fluorescence peaks have been measured at 655 and 685 nm respectively. These peaks are group-specific and changes in the intensity of fluorescence at peaks as well as the ratio of peaks are useful for determining the dominance and dynamics of change in the phytoplankton composition.
Comment 6: Cyanobacteria, green algae, and diatoms are mentioned in the manuscript, but it does not explain how to determine abundance through fluorescence spectra. More explanation is needed here.
Response 6: In the two estuaries taken in this study, Cyanobacteria and green algae are the dominant phytoplankton groups. We have taken the fluorescence peaks and fluorescence spectra to determine the dominance of these two groups and their proportionate presence in the estuarine water. Fluorescence peak at 655 nm with 570 nm excitation (excitation of phycocyanin) was the indicator of cyanobacteria and the fluorescence intensity was proportional to their density. Similarly, fluorescence peak at 685 nm with 440 nm excitation (Excitation of chlorophyll a) was the indicator of the dominance of green algae. The peaks and the shape of the spectra vary with the dominance of the phytoplankton group.
Comment 7: In line 287-288: To the best of our knowledge, this study reports for the first time the efficiency of Chl a fluorescence as a marker for predicting the air-water CO2 flux in estuaries. The authors only studied 18 stations in the Mahanadi and Dhamra estuaries in the Bay of Bengal, the amount of data is too small and not representative of all estuaries.
Response 7: In this study, we attempted to relate the fluorescence parameters to CO2 flux for the first time. This result is valid for regional scale flux studies and can be taken as a representative for tropical estuaries. However, this amount of data may fall short in predicting CO2 flux and extrapolating the results to global scale flux estimation.
Comment 8: Please unify the abbreviations in the manuscript.
Response 8: We have unified the abbreviations throughout the manuscript.
Comment 9: Please add p-value in all pictures. Moreover, the form of the picture is too simple.
Response 9: p-value is added to the respective figures. The intention of making box plots is to show the seasonal variation of physicochemical parameters clearly.

Round 2
Reviewer 2 Report
The authors said they computed the CO2 fugacity of water [fCO2 (water)] using CO2SYS.EXE software, but it’s common knowledge that there are four parameters in seawater carbonate system, namely pH, alkalinity, dissolved inorganic carbon (DIC) and pCO2 (or fCO2), and one has to use at least two parameters to calculate others. However, the authors only listed pH in the manuscript and did not refer to Alkalinity or DIC at all in the full text. It is impossible to calculate pCO2 (or fCO2) only using pH. Overall, I don’t think the data could support the results or the conclusions in the manuscript. In addition, the language should be cleaned up since there are many grammar problems and typos.
Author Response
Respected Reviewer,
You have correctly pointed out the issue and we completely agree with you in this regard. we have now included all the mentioned parameters, their measurement protocol, and results, in the revised manuscript. We sincerely hope that these inclusions to the revised manuscript eradicate all the confusion. We have removed all the grammatical errors and typos and improved it as much as possible. All the changes have been highlighted in the manuscript.
